# Impact of Processing and Preservation Methods and Storage on Total Phenolics, Flavonoids, and Antioxidant Activities of Okra (*Abelmoschus esculentus* L.)

**DOI:** 10.3390/foods12193711

**Published:** 2023-10-09

**Authors:** Maher M. Al-Dabbas, Majd Moumneh, Hani J. Hamad, Mahmoud Abughoush, Balkees Abuawad, Bha’a Aldin Al-Nawasrah, Rawan Al-Jaloudi, Sehar Iqbal

**Affiliations:** 1Department of Nutrition and Food Technology, Faculty of Agriculture, The University of Jordan, Amman 11942, Jordan; maher.dabbas@aau.ac.ae (M.M.A.-D.); majdmoumneh95@icloud.com (M.M.); dr.bhaa.alnawasrah@icloud.com (B.A.A.-N.); 2Nutrition and Dietetics Program, College of Pharmacy, Al Ain University, Abu Dhabi 112612, United Arab Emirates; mahmoud.abughoush@aau.ac.ae (M.A.); balkees.abuawad@aau.ac.ae (B.A.); 3Department of Clinical Nutrition and Dietetics, Faculty of Allied Medical Sciences, Philadelphia University, Amman 19392, Jordan; hhamad@philadelphia.edu.jo; 4Department of Clinical Nutrition and Dietetics, Faculty of Applied Medical Sciences, The Hashemite University, P.O. Box 330127, Zarqa 13133, Jordan; 5Faculty of Zarqa College, Al-Balqa Applied University, Zarqa 313, Jordan; rawanaljaloude@yahoo.com

**Keywords:** okra, phenolic, DPPH, preservation, antioxidant activity

## Abstract

Very few studies have thus far evaluated the impact of various processing and preservation techniques (blanching, frying, freezing, dehydration, and sun drying) on the levels of total phenolics, flavonoids, and antioxidant activities of okra. The primary objective of this study was to evaluate the effects of different processing and preservation methods on the levels of phenolics, flavonoids, and antioxidant activities of okra. The ethanolic extracts of each sample were analyzed before and after preservation and storage for a period of three months. The results showed a significant improvement (*p* < 0.05) in total phenolic content (134.1 mg GAE/100g) and DPPH (1-1-diphenyl1-2-pricrylhydrazyl) scavenging activity (IC_50_ value of 3.0 mg/mL) in blanched okra when compared to fresh okra (86.35 mg GAE/100g and IC_50_ value of 3.8 mg/mL, respectively). Fresh okra exhibited the highest flavonoid content (105.75 mg QE/100g), while sun-dried okra samples stored for three months exhibited a decrease in total phenolic content (14.45 mg GAE/100g), total flavonoid contents (13.25 mg QE/100g), reducing power activity (23.30%), and DPPH scavenging activity (IC_50_ value of 134.8 mg/mL). The DPPH inhibition activities of all okra treatments showed a significant and positive correlation with the okra phenolic and flavonoid content (r = 0.702 and 0.67, respectively). The reducing power activity (%) of okra treatments exhibited a strong correlation (r) with phenolic contents (r = 0.966), and the correlation with flavonoid contents was 0.459. Generally, different processing and preservation methods of okra revealed that the impact on total phenolic and flavonoid contents, as well as antioxidant activities, was slightly significant among samples preserved using the same method during storage. In addition, blanched and frozen okra resulted in the highest retention of phenolic contents and antioxidant activities.

## 1. Introduction

Okra (*Abelmoschus esculenntus* L.) is an important vegetable crop grown across most of the tropical and subtropical regions and even extending into Mediterranean climates. It belongs to the family *Malvaceae*, is commonly known as Lady’s Finger, is widely consumed as a vegetable, and is considered a staple food in many low-income, challenged nations. According to the USDA’s SR-21 dataset, the nutritional composition of okra comprises approximately 2% protein, 0.1% total fat, 7.0% carbohydrates, and 3.2% dietary fiber. Moreover, it is recognized as a promising reservoir of vitamins and mucilage, which are known as vital nutritional components of the diet [1,2,3].

Okra seeds have been utilized for their oil content, yielding varying concentrations from 20% to 40% of the total composition. Okra seed oil is notably abundant in linoleic acid, which is one of the essential fatty acids for nutritional requirements. Okra seeds are known as high-quality proteins, distinguished by their essential amino acid content, setting them apart from other plant-based protein sources [4].

Okra contains other valuable nutrients, such as a rich source of soluble fiber (gums and pectin). These soluble fibers have the potential to attenuate serum cholesterol levels and related adverse effects like heart disease. In addition, okra contains insoluble fiber, several carbohydrates, minerals, and vitamins, all of which contribute significantly to human health and overall well-being [5]. Earlier studies have shown a range of potential health benefits of okra for cardiovascular diseases, type 2 diabetes, digestive disorders, and even certain cancers [5,6,7,8].

Okra pods and seeds are rich in phenolic compounds, including quercetin and flavonol derivatives, catechin oligomers, and hydroxycinnamic derivatives (Figure 1). These compounds hold substantial biological significance in bestowing antioxidant activity in okra [5,9,10].

The polysaccharides present in okra exhibit a notable hypoglycemic effect, playing a role in the regulation of sugar absorption within the intestinal tract [11,12]. Furthermore, the consumption of okra contributes to the alleviation of ulcers and lung inflammation, lowers cholesterol levels in the blood, is beneficial for asthma patients with sore throats as well as irritable bowel conditions, and may have potential cancer-preventive properties [1,6,13,14]. These valuable attributes have led dietary experts to harness the benefits of okra in their recommendations [15].

Studies have shown that various preparation and processing techniques as well as preservation procedures might reduce the phytochemical constituents, nutritional value, and antioxidant properties of vegetables, including okra [1,16,17,18]. However, limited studies have been conducted on the impact of dehydration, freezing, and frying on the phenolic and antioxidant properties of okra. Therefore, this study aimed to assess the impact of different processing and preservation methods on the levels of phenolics, flavonoids, and antioxidant activities of okra.

## 2. Material and Methods

### 2.1. Chemicals

Gallic acid, aluminum trichloride (AlCl_3_), quercetin, and L-ascorbic acid were bought from Sigma-Aldrich (Steinheim, Germany). Folin-Ciocalteu reagent (FCR) and trichloroacetic acid were bought from AppliChem GmbH (Darmstadt, Germany). Ferric chloride (FeCl_3_), Di-Potassium hydrogen phosphate extra pure (K_2_HPO_4_·2H_2_O), sodium carbonate (Na_2_CO_3_), and potassium ferricyanide (K_3_[Fe(CN)_6_]_6_) were purchased from E. Merck (Darmstadt, Germany). Other chemicals were of reagent grade and purchased from local companies. Potassium dihydrogen phosphate (KH_2_PO_4_·2H_2_O) was purchased from Fluka-Garantie (Buchs, Switzerland). 1,1-diphenyl-2-picrylhydazyl (DPPH) was purchased from ICN Biomedicals Inc. (South Chillicothe Road Aurora, Aurora, OH, USA).

### 2.2. Sample Preparation

Freshly harvested okra was procured from a local market. After meticulous cleaning and the removal of inedible parts, the okra was subjected to the subsequent treatments:

The first part (200 g) was taken for immediate analysis of phenolic and antioxidant activities and served as the control sample, and the other part (200 g) was blanched at 80 °C for 5 min (B). The remaining quantity underwent preservation via the subsequent methods:(1)Freezing (BF): A batch of 500 g underwent blanching for a duration of 5 min, followed by gradual cooling to room temperature (25 °C), kept in plastic bags, and stored in a standard home freezer at −18 °C for 3 months.(2)Freezing (FF): Another 500 g batch was stir-fried with corn oil in a frying pan over medium-high heat (170 °C) for 5 min (F), when the okra became golden brown, it was removed. The fried okra was then drained on a paper towel-lined plate to facilitate the removal of excess absorbed oil, cooled at RT, and kept frozen in plastic bags at −18 °C in a normal home freezer for 3 months.(3)Sun drying (SD): A batch of 500 g of okra was spread out in a single layer on perforated stainless-steel trays and sun dried at 25–35 °C in an area that receives direct sunlight and good airflow for 3 months. The okra was regularly turned twice a day during the sun-drying process to ensure uniform drying.(4)Convection oven drying (OD): A batch of 500 g of okra was arranged on perforated stainless-steel trays and dried through the application of hot air in a conventional oven (Gallenkamp, UK) at 70 °C for 5 days. Following the drying period, the dehydrated okra was stored in a sealed glass jar at room temperature for 3 months.

The analysis of fresh and treated okra samples was conducted immediately after their respective processing stages. Additionally, evaluations were carried out monthly throughout the 3-month storage period. The moisture content (%) of fresh okra and other treatments was determined in duplicate using a conventional oven at 105 °C until a constant weight was achieved (Memmert, Model UFE500, Schwabach, Germany). This trial was conducted in triplicate for each individual sample.

### 2.3. Extraction of Okra Samples

The okra sample equivalent to 10.0g of dry weight bases (DWB) from each treatment was homogenized and extracted with 30 mL of hot absolute ethanol (96%) using a stainless steel electrical blender for 15 min. Afterwards, the mixture was filtered with Whatman No.1 filter paper to obtain extract I. The residues were re-extracted again with 30 mL of hot ethanol and filtered using Whatman No. 1 to obtain extract II. Both extracts (I and II) were combined, filtered again, and diluted with 100 mL of ethanol. The yield of okra extracts (mg/mL) was calculated. The obtained ethanol extracts of okra were then used to determine the total phenolic and flavonoid content, DPPH-radical scavenging activity, and reducing power activity. The residues were reextracted again by the same method with hot ethanol. The total extraction and blending take around 30 min. Hot ethanol is added to the extraction paragraph.

### 2.4. Determination of Total Phenolic Content

The total phenolic content of okra was evaluated by Folin-Ciocalteau reagent (FCR) [5]. Phenolic compounds were taken as mg gallic acid per 100 g of dry weight sample (mg GAE/100g) and were determined from regression equation based on the calibration curve: Y = 0.072x − 0.0356, R^2^ = 0.9844. Where x is the gallic acid concentration and Y is the absorbance. An aliquot of 0.1 mL from fresh or treated okra extracts was then poured into a 10 mL volumetric flask with the addition of 2.5 mL distilled water. Later, FCR (250 µL) was added and mixed thoroughly. Further, 0.5 mL of 10% (*w*/*v*) sodium carbonate was mixed after 3 min. After 1 h standing in a dark place, the absorbance at 760 nm was measured by spectrophotometer (model UVD-2900, Labomed, Los Angeles, CA, USA). All measurements were performed in triplicate.

### 2.5. Determination of Total Flavonoid Content

Flavonoids in fresh and okra treatments were assessed by the Miliuskas method [19]. For this purpose, 0.2 mL of ethanolic extracts from each treatment was mixed with 1 mL aluminum trichloride in water solution (20 g/L) and diluted with water to 25 mL. The absorption was measured at 415 nm after 40 min at 20 °C. Also, quercetin standard solutions were evaluated following the same procedure to establish standard curve. The amount of flavonoids in ethanolic extracts of the treatments was taken as quercetin equivalents per 100 g of dry weight sample (mg QE/100g) and was determined using regression equation based on the calibration curve: Y = 0.0933x − 0.0396, R^2^ = 0.987. Where x is quercetin concentration and Y is absorbance. Three measurements were taken for each procedure.

### 2.6. Antioxidant Activities

#### 2.6.1. DPPH Free Radical Scavenging Assay

DPPH (1,1-diphenyl-2-picrylhydrazyl) was performed to assess the free radical scavenging activity in okra ethanolic extracts using the method of Hatano [20]. Briefly, different concentrations of each treatment previously extracted with ethanol (10g/l00mL) were mixed with 1 mL of a methanolic solution of DPPH (6 × 10^−5^ M), and the absorbance of each extract, standard (200 ppm BHT), and control was measured at 517 nm after 30 min. The IC_50_ of the DPPH (concentration that produces a 50% reduction in the DPPH activity) for fresh okra and each treatment was determined and calculated using the following method:

DPPH radical scavenging (%) = control absorbance − sample absorbance/control absorbance × 100

#### 2.6.2. Reducing Power Activity (%)

The reducing powers of okra ethanolic extracts for fresh and treatments were determined using the Yildirm method [21]. An aliquot of 0.2 mL of each extract was mixed with 2.5 mL phosphate buffer (0.2 M, pH 6.6) and 2.5 mL (10 g/L) potassium ferricyanide. The mixture was then incubated at 50 °C for 30 min. An aliquot of 2.5 mL trichloroacetic acid (100 g/L) was then added, and the mixture was centrifuged at 1650× *g* for 10 min. Then 2.5 mL of upper layer solution was taken and mixed with 2.5 mL ferric chloride (1 g/L). The absorbance was measured at 700 nm for the ethanolic extracts and standard of ascorbic acid (30 µg).

### 2.7. Statistical Analysis

Complete Randomized Design (CRD) with three replicates was used as the design of the experiment. A statistical analysis was performed by SAS program, 2000 (SAS Institute Inc., Cary, NC, USA) for the results. Further, LSD test was applied to measure the difference between treatment means. A value of *p* < 0.05 was considered to show significant results.

## 3. Results and Discussion

### 3.1. Effects of Processing, Preservation Methods, and Storage on Total Phenolic and Flavonoid Contents of Okra

Table 1 shows the average total phenolic compound contents of fresh, blanched, frozen, fried, and dried okra samples after processing and during storage for three months under different conditions. Our results found that total phenolic and flavonoid contents were significantly affected by different treatment and storage conditions. Phenolics and flavonoids contents in fresh okra were 86.35 mg GAE and 105.1 mg QE/100g, respectively. The phenolic contents of okra were considerably increased after blanching (134.1 mg GAE/100g) and were higher as compared to the fresh sample (*p* < 0.05). This increase might be because of the efficient release of bound phenolics after the softening of okra tissues. However, flavonoid content was significantly (*p* < 0.05) decreased (60.4 mg QE/100g). This lower content may be due to the leaching of water-soluble flavonoids after tissue softening. These findings were similar to those reported by Turkmen et al. [20] and Randhir et al. [22], where an elevation in total phenolic content was observed in certain vegetables like peppers, broccoli, green beans, and corn after blanching, boiling, and steaming. Generally, thermal treatments exert a detrimental impact on flavonoid and phenolic compounds due to their inherent instability [15]. The difference in phenolics and flavonoids among various vegetables is due to the diverse phenolic groups present in the vegetables, thereby influencing their vulnerability to alteration or degradation during the cooking processes [3].

The phenolic content of blanched and frozen okra after one month of freeze storage was the highest (*p* < 0.05) among all treatments (167.1 mg GAE/100g). However, as the freeze storage continued in the second and third months, there was a significant decrease in the phenolic contents (156.15 and 148.70 mg GAE/100g, respectively). Flavonoids content was significantly decreased from 60.43 to 45.73 (mg QE/100g) after blanching and freeze storage for a month. On the same lines, there were no significant differences observed in flavonoid content between the second and third months of freeze storage (42.21 and 40.45 mg QE/100g, respectively).

The increase in phenolic contents and the decrease in flavonoids content after blanching and freeze storage might be due to the damage to okra cells caused by the growth of large ice crystals resulting during freeze storage, further development of texture softening and cell separation, and finally reduction of phytochemicals and antioxidant compounds [23]. However, freezing and storage of okra have been shown to be the gentlest methods of preservation for the total phenolic and flavonoid contents.

The frying process applied to okra (F) induced noteworthy alterations in the total phenolic and flavonoid contents when compared with fresh okra. The phenolic content of fresh okra decreased from 86.35 to 75.35 (mg GAE/100g), while flavonoids content was extensively decreased (*p* < 0.05) from 105.10 to 55.75 (mg QE/100g) after frying. The decrease in the levels of both total phenolic and flavonoid contents in vegetables after the frying process has been reported by several authors [20,23,24].

The reduction in phenolic content in fried vegetables has been attributed to several factors, such as the elevated frying temperature, enzymatic oxidation, leaching of some flavonoids from vegetables into cooking oil, and degradation during the frying process [25]. The phenolic content of fried and frozen okra during the first month of storage (FF1) decreased insignificantly from 75.35 to 72.25 (mg GAE/100g). However, a significant decline in total phenolic content was observed in the second (FF2) and third (FF3) months of freeze storage (56.75 and 53.3 mg/100g, respectively). The variation between values may be due to the cells being damaged during frozen storage.

The fried and frozen okra after one month of storage (FF1) is shown to have the highest total flavonoid contents (64.08 mg QE/100g) among other treatments, but significantly lower than the total flavonoid contents of the fresh sample (105.1 mg QE/100g). These results confirmed that the frozen product still has changes during freeze storage and can induce different changes affecting the bio-accessibility of phytochemical compounds as compared to fresh vegetables [26]. No significant differences were observed in the flavonoid contents of fried and frozen okra during storage.

As shown in Table 1, the total phenolic and flavonoid contents of samples changed remarkably after drying (*p* < 0.05) between sun SD and OD okra samples during storage. After 3 months of storage, the SD3 had the lowest total phenolic and flavonoids contents (14.45 mg GAE/100g and 13.23 mg QE/100g, respectively), while oven-dried okra and after three months of storage (OD3), the phenolic contents decreased from 44.6 to 18.8 (mg GAE/100g) and the flavonoids content decreased from 39.3 to 25.5 (mg QE/100g). Moreover, the storage period of the same drying method slightly affected the phenolic and flavonoid contents. For example, in SD, the phenolic content decreased from 15.53 to 14.45 (mg GAE/100g) after three months of storage, while the flavonoids content significantly decreased (*p* < 0.05) from 28.23 to 13.23 (mg QE/100g) for the same period. Degradation of phenolic compounds by drying, polyphenol oxidation by oxygen, or the use of phenolic compounds as reactants in the Maillard reaction during the drying process could be the possible reason for a reduction in the total phenolic contents of dried okra [27,28,29]. Moreover, the binding of polyphenols with other compounds modifies their chemical structure due to heat treatment [30,31]. The lowest total flavonoid content might be attributed to polymerization during air-drying [13] or harsh conditions during the drying process that can affect cell wall integrity, leading to the reduction of flavonoid content [32,33].

However, the general trend shows that the effect of storage period on phenolic and flavonoid content was slightly significant among samples dried by the same method but significant for samples dried by different methods.

### 3.2. Effects of Processing, Preservation Methods, and Storage on Antioxidant Activity of Okra

#### 3.2.1. DPPH Free Radical Scavenging Activity

Table 1 shows significant (*p* < 0.05) effects of different processing, preservation methods, and storage conditions over three months on DPPH radical scavenging activity evaluated by IC_50_.

DPPH activity was significantly the highest in fresh, blanched, and frozen okra (BF) and the lowest in sun-dried okra (SD3) (IC_50_ = 134.0 mg/mL) after three months of storage, which appears to reflect the poor antioxidant activity of dried okra against DPPH free radicals among all treatments.

The fried okra (F) sample showed a significant decrease in DPPH inhibition activity (IC_50_ = 21.1 mg/mL); this decrease may be due to enzymatic oxidation and destruction of polyphenols and other antioxidant compounds during the preparation process or long period of frying [25]. However, antioxidant potential increased for fried and frozen okra (FF) during storage without significant differences, suggesting a positive effect of freezing on the release of bound phenolics for higher DPPH activity during freeze storage.

Dried okra samples showed the lowest values for DPPH free radical scavenging activities, and the DPPH activity considerably decreased during storage for a period of three months. When SD was compared with OD, the DPPH activity notably decreased with storage period, from an IC_50_ of 35.8 to 134.0 mg/mL after three months of storage, while the OD activity decreased from an IC_50_ of 50.4 to 112.1 mg/mL. This reduction in antioxidant activity might be related to total phenolic content losses as a result of the oxidation and polymerization of phenolic compounds, which are the main compounds responsible for the antioxidant activity of plants [2,28]. It was reported that DPPH inhibition was the most affected by processing techniques, especially dehydration, which is considered the most detrimental preservation technique [34,35]. The DPPH radical scavenging activity (%) for the treatments showed a strong correlation with the flavonoid content (r = 0.672). With phenolic content, the correlation (r) was 0.702. The phenolic and flavonoid contents exhibited excellent and positive associations with DPPH scavenging activities for different treatments during storage, suggesting their ability to neutralize DPPH radicals, and both treatment and storage periods highly affected the bioactive compounds of okra. These results are in agreement with many previous findings showing the significant effect of available bioactive phenolic compounds in vegetables on their antioxidant properties [10,18,34,36].

#### 3.2.2. Reducing Power Activity (%)

The reducing power means the ability of the antioxidant to reduce Fe^+3^ to Fe^+2^. In this regard, Figure 2 shows the effect of processing, preservation methods, and storage for three months on reducing the power activity (%) of okra. The reducing power activity of vitamin C (30 µg) was considered 100%, although there were no remarkable differences in reducing power activity between fresh okra (99.1%). Furthermore, compared to the standard vitamin C, the reducing power activity of okra after blanching was significantly (*p* < 0.05) increased (145.8%) and was higher than that of a fresh sample because of the efficient release of bound phenolics due to the softening of okra tissues.

The blanched and frozen okra (BF1) showed the highest and most significant reducing power activity (174.0%) after 1 month of storage, indicating a strong antioxidant activity comparable to the standard ascorbic acid. However, the reducing power activity of the blanched and frozen okra (BF2 and BF3) did not decrease significantly (161.7% and 157.3%) after two and three months of storage, which might be due to enzymatic reactions in frozen products and its slow activity in the frozen state [11]. The reducing power activity (%) for fried (96.9%) and fried frozen okra (FF1, FF2, and FF3) was slightly reduced during freeze storage (93.6%, 92.3%, and 90.4%, respectively), and when compared to the fresh or fried okra reducing power activity, the results were not significant.

Figure 2 shows the substantial effect of different drying and storage conditions on reducing power activity compared with other processing and preservation methods; however, the dried okra samples did not show remarkable differences in measured reducing power activity during storage conditions as a result of the effects of light (sun drying) or heat (oven drying).

The reducing power activity (%) for sun-dried okra during storage (SD1, SD2, and SD3) was 24.6%, 23.7%, and 23.5%, respectively. For oven-dried okra, OD1 and OD3, the reducing power activities were 21.1% and 19.6%, respectively. The reduction in reducing power activities of dried okra during storage may suggest a reduction in the quality of phenolic and flavonoid compounds due to an increase in oxidative degradation under high temperatures or light.

The reducing power activity (expressed as 30 μg vitamin C equivalent) (%) for the treatments showed a slight positive correlation with the flavonoids content (r = 0.459), while total phenolic contents presented a positive correlation (r = 0.966). This may suggest that reducing power activity is more highly affected by the phenolic content after processing and preservation than by the contents of flavonoids. Previously, a positive correlation between polyphenol contents and reducing power activity has been reported for fruits and vegetables in various studies [2,35].

Since various contradictory information is available in the literature regarding the effect of various processing techniques on bioactive compounds during storage, this study will help to make consumers aware of the best processing, cooking, and preservation techniques to reduce the degradation of some bioactive compounds and thus make them more suitable for traditional medicine. Also, the studied methods are commonly used to preserve okra, either on a home or commercial scale. Our results will provide essential information for consumers to select the best processing and preservation technique for okra.

## 4. Conclusions

The effect of processing and preservation techniques on okra during storage varies significantly. Our results indicate a substantial (*p* < 0.05) influence of processing, preservation methods, and storage conditions on the antioxidant activity, phenolic, and flavonoid contents of okra. Blanching and freezing of okra resulted in elevated levels of phenolic and flavonoid contents as well as enhanced antioxidant activities, whereas dried okra exhibited lower levels. Drying okra resulted in the lowest retention of phenolic, flavonoid, and antioxidant activities, while freezing resulted in the highest retention of phenolic, flavonoid, and antioxidant contents. However, a statistically significant (*p* < 0.05) difference was observed when comparing samples processed and preserved using different methods. Overall, different processing and preservation methods applied to okra show a slight effect on total phenolic and flavonoid contents and antioxidant activities within samples preserved using the same method throughout storage.

In conclusion, our results indicate that blanching for a short time followed by freezing is the best method of preservation and storage in terms of phenolic, flavonoids, and antioxidant activity retention, while drying, especially sun drying, was the least effective. This may suggest that blanching followed by freezing is the best method for okra preservation.

## Figures and Tables

**Figure 1 foods-12-03711-f001:**
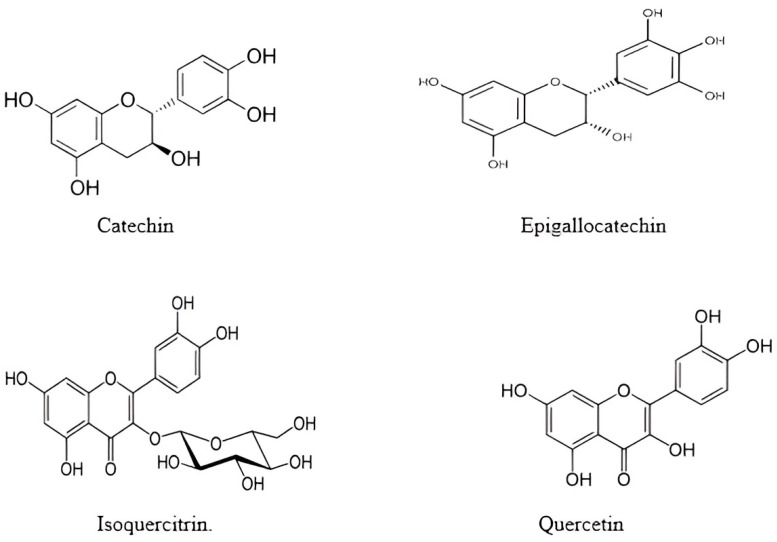
Chemical structures of some phenolic compounds derived from okra.

**Figure 2 foods-12-03711-f002:**
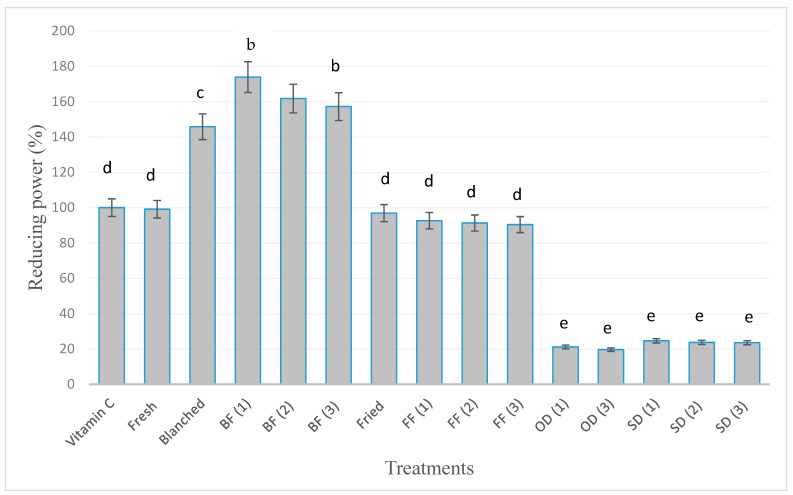
Means of reducing power (%) ± standard deviation. Values with different letters are significantly different (*p* ≤ 0.05). BF(1) (blanched and frozen for 1 month), BF(2) (blanched and frozen for 2 months), and BF(3) (blanched and frozen for 3 months), FF(1) (fried and frozen for 1 month), FF(2) (fried and frozen for 2 months), and FF(3) (fried and frozen for 3 months), SD(1) (sun dried and stored for 1 month), SD(2) (sun dried and stored for 2 months), and SD(3) (sun dried and stored for 3 months), and OD(1) (oven dried and stored for 1 month) and OD(3) (oven dried and stored for 3 months).

**Table 1 foods-12-03711-t001:** Average total phenolic (mg GAE/100g) and flavonoid (mg QE/100g) contents and DPPH scavenging activity (IC_50_ (mg/mL)) of fresh, blanched, fried, frozen, and dried okra treatments *.

Okra Treatments Extract	Phenolics (mg GAE/100g)	Flavonoids (mg QE/100g)	DPPH IC_50_ (mg/mL)
Fresh okra	86.35 ± 0.35 ^e^	105.10 ± 0.82 ^a^	3.81 ± 0.85 ^h^
Blanched okra (B)	134.10 ± 1.42 ^d^	60.43 ± 0.97 ^b,c^	3.02 ± 0.17 ^h^
(BF1)-Frozen (1 month)	167.10 ± 2.10 ^a^	45.73 ± 0.52 ^e^	2.91 ± 0.34 ^h^
(BF2)-Frozen (2 months)	156.15 ± 3.95 ^b^	42.21 ± 0.85 ^f^	3.06 ± 0.15 ^h^
(BF3)-Frozen (3 months)	148.70 ± 4.28 ^b,c^	40.45 ± 0.73 ^f^	3.65 ± 0.37 ^h^
Fried okra (F)	75.35 ± 1.05 ^f^	55.75 ± 0.17 ^d^	21.1 ± 0.46 ^f^
(FF1)-Frozen (1 month)	72.25 ± 2.65 ^f^	64.08 ± 1.78 ^b^	12.07 ± 0.59 ^g^
(FF2)-Frozen (2 months)	56.75 ± 2.65 ^g^	62.05 ± 1.35 ^b^	12.84 ± 0.64 ^g^
(FF3)-Frozen (3 months)	53.30 ± 1.28 ^g^	60.42 ± 0.90 ^b,c^	12.91 ± 0.72 ^g^
(SD1)-Sun drying (1month)	15.53 ± 1.24 ^j^	28.23 ± 0.64 ^g^	35.8 ± 0.52 ^e^
(SD2)-Sun drying (2 months)	14.70 ± 1.14 ^j^	13.85 ± 0.26 ^i^	57.8 ± 0.40 ^c^
(SD3)-Sun drying (3 months)	14.45 ± 2.36^j^	13.23 ± 0.52 ^i^	134.0 ± 0.34 ^a^
(OD1)-Oven drying (1month)	44.60 ± 1.55 ^h^	39.31 ± 0.37 ^f^	50.4 ± 0.29 ^d^
(OD3)-Oven drying (3 months)	18.80 ± 0.99 ^i^	25.57 ± 0.62 ^h^	112.1 ± 0.62 ^b^

* Means of phenolic and flavonoids contents and DPPH (IC_50_) ± standard deviation. Values with different letters are significantly different (*p* ≤ 0.05). BF1 (blanched and frozen for 1 month), BF2 (blanched and frozen for 2 months), and BF3 (blanched and frozen for 3 months), FF1 (fried and frozen for 1 month), FF2 (fried and frozen for 2 months), and FF3 (fried and frozen for 3 months), SD1 (sun dried and stored for 1 month), SD2 (sun dried and stored for 2 months), and SD3 (sun dried and stored for 3 months), and OD1 (oven dried and stored for 1 month) and OD3 (oven dried and stored for 3 months).

## Data Availability

The data used to support the findings of this study can be made available by the corresponding author upon request.

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
