# Peer review of "Impact of Processing and Preservation Methods and Storage on Total Phenolics, Flavonoids, and Antioxidant Activities of Okra (Abelmoschus esculentus L.)"

_foods, 2023, doi:10.3390/foods12193711_

Round 1

Reviewer 1 Report

Comments and Suggestions for Authors

The manuscript presents a comprehensive study on the effects of various processing and preservation methods on the phenolic, flavonoid contents, and antioxidant activities of okra. The article is fairly well written. My comments for further improvement:

  1. The Results and Discussion section is dense and could benefit from further subdivision into more specific subsections to improve readability and comprehension. For example, you could separate the findings related to phenolic and flavonoid contents from those concerning antioxidant activities. Each of these could then be further broken down based on the different processing and preservation methods you studied blanching, freezing, frying, and drying. This would allow readers to follow the flow of your research and understand the impact of each method on the different variables you measured more easily.
  2. While the manuscript is thorough, it would be beneficial to clearly state the novelty of this study in comparison to existing literature. What makes your approach or findings unique? Truly speaking I don’t see any novelty in the manuscript. Please justify.
  3. While you mention correlations between phenolic and flavonoid contents and antioxidant activities, a more in-depth discussion on the implications of these correlations would add value. Here you can bring some novelty. Correlation is not new; everyone is aware of it.
  4. Ensure that units are consistently presented throughout the manuscript to avoid any confusion. For example, if you're using milligrams per 100 grams (mg/100g) to express the concentration of phenolic compounds in one section, it's important to use the same units when discussing flavonoid concentrations or any other similar measurements. Switching between different units like mg/100g, mg/kg, or µg/g can confuse the reader and make it difficult to compare results directly.
  5. The conclusion section could be expanded to summarize the key findings more explicitly and to suggest potential applications or future research directions.
  6. Are there any specific reasons for choosing the processing and preservation methods that you studied? What are the practical implications of your findings, especially in the context of food preservation and health benefits?
Comments on the Quality of English Language

Reads well.

Author Response

Title; Impact of processing and preservation methods and storage on total phenolics, flavonoids and antioxidant activities of Okra (Abelmoschus esculentus L.)

Submission ID: 2626445

Dear Editor, Dear Reviewers,

We would like to thank the reviewers for their in-depth analysis and suggestions to improve our manuscript. We addressed all points and revised the script accordingly. Revisions are marked in yellow in the script.

We performed revisions especially i) Improved methodology, results, and discussion according to the suggestions, ii) Added study strengths and novelty iii) revised conclusion and research implications. Furthermore, all criticized points were addressed and suggestions from the reviewers were incorporated in the latest manuscript.

We hope that this revision is satisfactory. Thank you.

Reviewers 1

N0.

Comments

Response

1-

Dense of results and discussion part and subdivision to subsections.

·         Thank you for your comment. The results and discussion in this article aimed to make comparison between different treatments and their impact on phenolics, flavonoids and antioxidant activity (DPPH) during storage period. Subsections for each treatment during storage may not show the impact of treatments and require a new statistical analysis for each treatment during storage.  

2-

Novelty of the study

·         Thank you for your valuable suggestion. We have now added the novelty of the study in the revised manuscript. The changes are mentioned at Page 9, Line 362-367.

3-

More in-depth discussion on the implications of correlations between phenolic and flavonoid contents and antioxidant activities.

·         We have now revised the discussion section and added the related information at 309-312, and from 357-361 for more explanation of correlations

4-

Ensure that units are consistently presented throughout the manuscript to avoid any confusion. For example, if you're using milligrams per 100 grams (mg/100g) to express the concentration of phenolic compounds in one section, it's important to use the same units when discussing flavonoid concentrations or any other similar measurements. Switching between different units like mg/100g, mg/kg, or µg/g can confuse the reader and make it difficult to compare results directly.

·         We have completely checked the revised manuscript and made appropriate changes if required through the manuscript

5

The conclusion section could be expanded to summarize the key findings more explicitly and to suggest potential applications or future research directions.

·         Conclusion section is revised according to the suggestions.

6.

Are there any specific reasons for choosing the processing and preservation methods that you studied? What are the practical implications of your findings, especially in the context of food preservation and health benefits?

·         These methods that are commonly used to preserve okra either in home scale or commercial scale. We have added the practical implication in the end of discussion section.

Reviewer 2 Report

Comments and Suggestions for Authors

Dear Editors and authors,

this is intersting piece of research and needs some extensions and modifications.

Abstract: It is okay that authors compared methods for preservation of okra however I miss the clear aim of the research and it is not outlined in the abstract.  Moreover: What is the importance and application of the results? Methods differ this is okay but what is the benefice of it to science and industry?

L43: This sentence is saying nothing at all.

Please use Latin names at the first mention of the Species.  It is not necessary to use Latin names again and again throughout of the manuscript.

Figure 1:  the structures of the compounds are distorted.  Please correct. 

Section 2.2.:  method description for preservation are ambiguous and not consistent with table 1 abbreviations.  Please use uniform abbreviations.  Method A is actually not freezing it is blanching.  Method B is frying and not freezing.  Indicating freezing is confusing  for both methods.  Freezing is regarded more as a storage method.  Why is sun-drying and convection drying indicated with 2 and 3 as methods? 

Section: 2.3.  The most crucial part of the manuscript and the research is the measurement of the water content.  Authors state that for extraction 10 gram of dry okra was taken.  As with preservation and storage the moisture content changes significantly it is crucial to monitor the water content.  However authors do not describe method for this. Please indicate how you measured the water content. Did you measure water content before each extraction step? How did you assure that 10 gram equivalents on dry weight base were really tested?

Are all results given 100 gram dry weight basis?  If so there is no need to indicate it every time  because it is disturbing. Moreover in L199  you indicate 64.0 mg QE/100 g  without mentioning dry weight basis. Is that OK?

L209: "was remarkably higher" I think it is better to write "was the highest"

Table1:  abbreviation method descriptions  or not caused this tent with section 2.2. What does DMB mean here? If all values were caluculated on dry mass basis there is need to mention this. 

However DPPH was determined for the extract and not related to dry mass.

The same applies to Figure 2 on abbreviations. In figure 2 caption abbreviations must be explained.

Please check is P<0.05 or p>0.05 are OK, becuase both of them appear in text. 

Some formulations is English are not OK, please check with a native Engish speaker.

Comments on the Quality of English Language

English should be checked by  a native speaker.

Author Response

Reviewer 2

Title; Impact of processing and preservation methods and storage on total phenolics, flavonoids and antioxidant activities of Okra (Abelmoschus esculentus L.)

Submission ID: 2626445

Dear Editor, Dear Reviewers,

We would like to thank the reviewers for their in-depth analysis and suggestions to improve our manuscript. We addressed all points and revised the script accordingly. Revisions are marked in yellow in the script.

We performed revisions especially i) Improved methodology, results, and discussion according to the suggestions, ii) Added study strengths and novelty iii) revised conclusion and research implications. Furthermore, all criticized points were addressed and suggestions from the reviewers were incorporated in the latest manuscript.

We hope that this revision is satisfactory. Thank you.

Reviewer 2

No.

Comments

Response

1.

Abstract: It is okay that authors compared methods for preservation of okra however I miss the clear aim of the research and it is not outlined in the abstract.  Moreover: What is the importance and application of the results? Methods differ this is okay but what is the benefice of it to science and industry?

·         Thank you for the recommendation, we have added the study objective in the abstract. Also, aim of research is explained at the end of introduction.

·         Regarding the application of the results, we have added following information before the conclusion section.

·         Since various contradictory information is available in the literature regarding the effect of various processing techniques on bioactive compounds during storage. This study will help to make consumers aware about the best processing, cooking and preservation techniques to reduce the degradation of some bioactive compounds and thus make it more suitable for traditional medicine. Also, the studied methods that are commonly used to preserve okra either in home scale or commercial scale. Our results will provide essential information for consumers to select the best processing and preservation technique of okra.

2

L43: This sentence is saying nothing at all.

·         We have removed the sentence.  

3.

Please use Latin names at the first mention of the Species.  It is not necessary to use Latin names again and again throughout of the manuscript.

·         Thank you for your suggestion, we have now removed the repetition of the Latin name.

4.

Figure 1:  the structures of the compounds are distorted.  Please correct.

·         We have checked the compounds’ structure and they are correct. However, the changes might be due to the conversion of word to pdf file. Therefore, we have attached compounds’ structure as a supplementary file as well.

5.

Section 2.2.:  method description for preservation are ambiguous and not consistent with table 1 abbreviations.  Please use uniform abbreviations.  Method A is actually not freezing it is blanching.  Method B is frying and not freezing.  Indicating freezing is confusing for both methods.  Freezing is regarded more as a storage method.  Why is sun-drying and convection drying indicated with 2 and 3 as methods?

·         We have completely checked the revised. Abbreviation adjusted to be uniform.

·         We have added text and relevant details and corrections for more clarification (Line 106-125).

Usually, we blanch vegetables before freezing to inactivate PPO enzymes and this may be accomplished by heating for several mints or for okra it is stir-fried before freezing for the same purpose.

-          They are different method in drying it take 5- 7 days to be dried but in sun drying more than 15 days were requested to accomplish the drying process.

6.

Section: 2.3.  The most crucial part of the manuscript and the research is the measurement of the water content.  Authors state that for extraction 10 gram of dry okra was taken.  As with preservation and storage the moisture content changes significantly it is crucial to monitor the water content.  However, authors do not describe method for this. Please indicate how you measured the water content. Did you measure water content before each extraction step? How did you assure that 10-gram equivalents on dry weight base were really tested?

·         It was an equivalent to 10.0 gram of okra was taken on DMB for each treatment and this done by moisture determination before analysis. A new clarification about moisture content is added lines (127-132).

-          For example: in fresh okra the moisture content was 82%, so the amount of okra that is equivalent to 10 grams was equal to: 10 X 1000/ 18 = 55.56 gram was taken for extraction and so on. So before extraction we have to determine moisture content to take an amount equivalent to 10 gram on dry bases.

7.

Are all results given 100-gram dry weight basis?  If so, there is no need to indicate it every time because it is disturbing. Moreover, in L199 you indicate 64.0 mg QE/100 g without mentioning dry weight basis. Is that, OK?

·         Yes, we have now adjusted all the text. Now all consistent 

8.

L209: "was remarkably higher" I think it is better to write "was the highest"

·         Corrected according to the suggestion

9.

Table1:  abbreviation method descriptions or not caused this tent with section 2.2. What does DMB mean here? If all values were caluculated on dry mass basis there is need to mention this.

However DPPH was determined for the extract and not related to dry mass.

The same applies to Figure 2 on abbreviations. In figure 2 caption abbreviations must be explained

·         Thank you for your detailed input. We have corrected. But the same method of taking a sample that is equivalent to 10 g on DWB then extracted with ethanol for studying.

·         DPPH and reducing power activity.

·         Moreover, abbreviation of figure 2 explained and corrected   

10.

Please check is P<0.05 or p>0.05 are OK, becuase both of them appear in text.

·         We have incorporated the changes.

11.

Some formulations are English are not OK, please check with a native Engish speaker.

·         Thank you for your input, we have revised the manuscript according to the suggestion.

Reviewer 3 Report

Comments and Suggestions for Authors

Autorzy pracy pt. „Wpływ metod przetwarzania, konserwacji i przechowywania na zawartość fenoli ogółem, flawonoidów i aktywność przeciwutleniającą okry 3 (Abelmoschus esculentus L.)” przedstawili zmiany w zawartości i aktywności przeciwutleniającej okry poddawanej różnym zabiegom technologicznym. Stosowane metody badawcze są obecnie traktowane jako uzupełniające, a nie główne ze względu na ich niedoskonałości związane z niskim współczynnikiem powtarzalności. Uważam, że publikacja ta jest zbyt niska w stosunku do przyjętego standardu badań publikowanych w tym czasopiśmie. Jeśli chodzi o komentarze do tekstu, nie ma ich wiele, najważniejsze to:

The extraction method used is rarely used due to its low extractivity, it is usually supported by ultrasound, or it is optionally added in this form, but the solvent penetration time is extended to 12 to 24 hours. The authors did not provide the extraction time in their work.

Line 107, no indication of the blanching method, water blanching or steam blanching

Line 130, the authors provide extraction with 100% ethanol, we are unlikely to encounter such pure ethanol, and much less concentrated ethanol is used for extraction. According to numerous publications, the most effective extraction is in solutions with concentrations ranging from 50 to 70%.

Line 140 lacks FCR abbreviation expansion

Author Response

Reviewer 3

Title; Impact of processing and preservation methods and storage on total phenolics, flavonoids and antioxidant activities of Okra (Abelmoschus esculentus L.)

Submission ID: 2626445

Dear Editor, Dear Reviewers,

We would like to thank the reviewers for their in-depth analysis and suggestions to improve our manuscript. We addressed all points and revised the script accordingly. Revisions are marked in yellow in the script.

We performed revisions especially i) Improved methodology, results, and discussion according to the suggestions, ii) Added study strengths and novelty iii) revised conclusion and research implications. Furthermore, all criticized points were addressed and suggestions from the reviewers were incorporated in the latest manuscript.

We hope that this revision is satisfactory. Thank you.

Reviewer 3

No

Comments

Responses

1.

The extraction method used is rarely used due to its low extractivity, it is usually supported by ultrasound, or it is optionally added in this form, but the solvent penetration time is extended to 12 to 24 hours. The authors did not provide the extraction time in their work.

·         Thank you for your in-depth analysis. We have the following information in the extraction section.

-          The okra pods were extracted using high-speed stainless-steel blender for 15 minutes and filtered. The residues reextracted again by the same method with hot ethanol.  So, the total extraction and blending is around 30 minutes. A hot ethanol is added to the extraction paragraph.

2.

 Line 107, no indication of the blanching method, water blanching or steam blanching

·         The blanching method is clarified by adding new sentence (106-107)

3.

 Line 130, the authors provide extraction with 100% ethanol, we are unlikely to encounter such pure ethanol, and much less concentrated ethanol is used for extraction. According to numerous publications, the most effective extraction is in solutions with concentrations ranging from 50 to 70%.

·         You are right, 50, 70, 96% (absolute ethanol) is usually used for extraction of bioactive compounds from plants and herbs. We used absolute ethanol which is (96%) and it is corrected in the manuscript (line 136). however, 50-70% hot ethanol extraction may involve the extraction of soluble sugars from okra unlike herbs. The objective was to extract bioactive materials but not sugars since the amount of dried matter in okra extracts are encountered in calculation.

4

Line 140 lacks FCR abbreviation expansion

·         It is mentioned at and written as Folin-Ciocalteau reagent (FCR) lines (145-146)

Round 2

Reviewer 3 Report

Comments and Suggestions for Authors

no coment